# Can rapid approaches to qualitative analysis deliver timely, valid findings to clinical leaders? A mixed methods study comparing rapid and thematic analysis

Beck Taylor, Catherine Henshall, Sara Kenyon, Ian Litchfield, Sheila Greenfield

Institute of Applied Health Research, College of Medical and Dental Sciences, University of Birmingham, Birmingham, UK

**Correspondence to**
Dr Ian Litchfield;
i.litchfield@bham.ac.uk

## ABSTRACT

**Objectives** This study compares rapid and traditional analyses of a UK health service evaluation dataset to explore differences in researcher time and consistency of outputs.

**Design** Mixed methods study, quantitatively and qualitatively comparing qualitative methods.

**Setting** Data from a home birth service evaluation study in a hospital in the English National Health Service, which took place between October and December 2014. Two research teams independently analysed focus group and interview transcript data: one team used a thematic analysis approach using the framework method, and the second used rapid analysis.

**Participants** Home birth midwives (6), midwifery support workers (4), commissioners (4), managers (6), and community midwives (12) and a patient representative (1) participated in the original study.

**Primary outcome measures** Time taken to complete analysis in person hours; analysis findings and recommendations matched, partially matched or not matched across the two teams.

**Results** Rapid analysis data management took less time than thematic analysis (43 hours vs 116.5 hours). Rapid analysis took 100 hours, and thematic analysis took 126.5 hours in total, with interpretation and write up taking much longer in the rapid analysis (52 hours vs 8 hours). Rapid analysis findings overlapped with 79% of thematic analysis findings, and thematic analysis overlapped with 63% of the rapid analysis findings. Rapid analysis recommendations overlapped with 55% of those from the thematic analysis, and thematic analysis overlapped with 59% of the rapid analysis recommendations.

**Conclusions** Rapid analysis delivered a modest time saving. Excessive time to interpret data in rapid analysis in this study may be due to differences between research teams. There was overlap in outputs between approaches, more in findings than recommendations. Rapid analysis may have the potential to deliver valid, timely findings while taking less time. We recommend further comparisons using additional data sets with more similar research teams.

**Strengths and limitations of this study**

► Our study explores a strategy to address the time-lag in reporting qualitative findings to clinicians and policymakers, which slows translation of research into practice.

► This is the first comparison of qualitative analytical methods in applied health research which compares both researcher time *and* outputs, with a complete study dataset.

► The work describes the process of comparing time and analytical outputs in detail, to inform others planning further methodological comparisons.

► Due to the time lag in thematic analysis outputs, our study did not triangulate findings with the original participants.

► The study uncovered important challenges in comparing analytical approaches between research teams which can inform the design future work in this area.

approaches.[1] Applications of qualitative methods include: early work to identify areas for focus; throughout a study to explore processes and user experience; and following a trial or intervention implementation to explain outcomes and/or identify stakeholder experiences, to explore in more depth questions or issues identified through quantitative work and to problematise or 'unpack' issues or topics taken for granted.[2] Increasingly this type of research can include a broader range of contributors, for example, where members of the public, patients, clinicians and researchers are involved in analysing and interpreting data to ensure a multidisciplinary perspective or pragmatically using several researchers to code data in the interests of time.[3 4]

Typically stakeholders want rapid results,[5–7] yet compared with quantitative approaches, traditional qualitative approaches often considerable time is required to manage and interpret data and deliver findings.[8 9] In

## BACKGROUND

Applied health research frequently adopts mixed methods, often using qualitative

a service context, delays may render the findings out of date, reducing their applicability and relevance. There are examples of apparently more rapid alternatives to traditional qualitative approaches, including specific end-to-end approaches such as Rapid Assessment Process and Rapid Ethnography.[6 9–13] There are four broad areas where time can be saved: by reducing data collection time, for example, by allowing less time between data collection episodes[6]; by reducing data management time, for example, by relying on untranscribed audio recordings, notes, summaries and mind maps[10–12]; by minimising the time spent analysing data by summarising as opposed to formally coding[11 13]; and by limiting the time spent on analysis by using a 'one sheet of paper' summary to explore a sample of a large precoded dataset.[9] Often rapid methods describe a broad approach, including activities from entering the field through to delivery of findings and/or involve mixed methods.[6 7] This paper specifically explores whether rapid *analysis* (RA) of qualitative data (distinct from end-to-end rapid methods) delivers equivalent findings to traditional approaches and how much time may be saved in practice.

There are a limited number of studies that have compared different qualitative analytical techniques.[11 14–16] In some of the empirical examples identified, methodologists have predominantly compared methods of data collection (eg, interviews vs internet forums[14]) and focused on the number and content of codes rather than interpretation. Of the three examples identified that compare analytical approaches, one used focus group data to compare thematic analysis (TA) of a partial dataset with mind-mapping of a full dataset.[11] While this paper provides minimal detail regarding the method of comparison, it reported differences in time taken to analyse the data and in the number and presentation of codes. The second example compared software-assisted and constant comparative approaches to analysis describing differences in the frequency of codes and coding levels.[15] The third example compared analysis of focus group data directly from audio recordings, with TA of transcribed data, and found that themes generated were comparable.[16]

The work we present here was conducted as part of the Collaboration for Leadership in Applied Health Research and Care (CLAHRC) programme in the West Midlands of England. CLAHRC involves local teams across universities and National Health Service organisations working in partnership to deliver research to improve services for patient benefit.[17] As part of a service evaluation study of a new home birth service, we gathered interview, focus group and documentary data. We then compared the speed and outputs of rapid and traditional techniques applied to the same dataset. For the RA, we used the approach developed by Hamilton at UCLA.[13] We compared this with TA[18] and the framework method, which was selected due to the team's existing familiarity with this approach and the fact that it is increasingly applied in multidisciplinary health services research.[8 19]

## METHOD
This study compares rapid and traditional analyses of a UK health service evaluation dataset to explore differences in researcher time and consistency of outputs. This was a mixed methods study, quantitatively and qualitatively comparing the outputs of qualitative methods.

## SETTING
The data came from a home birth service evaluation study in a hospital in the English National Health Service, which took place between October and December 2014. This was a service innovation put into place by the hospital. A dedicated team of midwives was set up to provide antenatal, birth and postnatal care to women choosing to have a home birth, with the aim of providing a more reliable service and increasing the local home birth rate.

### Characteristics of participants
Home birth midwives (6), midwifery support workers (4), commissioners (4), managers (6) and community midwives (12) and a patient representative (1) participated in the original study.

### Description of processes, interventions and comparisons
In the original evaluation, an evaluability assessment approach was adopted,[20] and its specific objectives were to: establish the original programme design and how the service differed from this design and why; identify facilitators or barriers to implementation; establish what service data are available and how it is being/could be gathered; and identify how staff would develop/improve the service. The evaluation was a qualitative study, involving interviews and focus groups with key participants involved in the home birth service.

Twenty-three provider and commissioning staff and one patient representative were purposively sampled, with recruitment by direct email or telephone invite, with three unable to take part due to availability. Twenty-one semistructured interviews informed by the study objectives were conducted by BT at participants' workplaces. A single focus group of 12 midwives was facilitated by BT and CH also structured according to the study objectives. A convenience sampling approach was taken for the focus group, with midwives available at the allotted time invited to take part at their workplace. Participants were not known to researchers prior to the study. Interviews and the focus group lasted approximately 1 hour, were digitally recorded and transcribed for analysis, with minimal field notes taken. Participants did not review transcripts. Eight key service documents were also used in the analysis (business case, service guidelines and commissioning policy). Local approval was obtained from the hospital research and development team. The data were analysed independently using first RA and second TA as described in detail below. All researchers work in applied health research in the same department of a UK university. BT is a public health physician, CH is a registered nurse and

SK is a registered midwife. Researchers 4 and 5 are health service researchers, and researcher 6 is a medical sociologist. A summary and comparison of the process used for the two analyses is shown in table 1. The work was undertaken using a theoretically interpretive, generic qualitative approach across both teams.

### The primary RA

RA was conducted between November and December 2014; this constituted the primary empirical work that was subsequently reported to the service. The rapid qualitative analysis approach used[13] was designed to deliver timely findings with methodological rigour. The approach includes guidance on data collection and report writing and was developed using teams of less experienced researchers. Here we have used only the analytical methodology and researchers experienced in qualitative methods. Hamilton relates how the reduced timeframe of rapid methods means that they tend to be more deductive and explanatory than inductive and exploratory.[13] It can be hypothesised that this may negatively impact on the ability of rapid methods to discover more 'hidden' phenomena that one associates with traditional qualitative methods, and this must be balanced with the speed at which rapid methods can deliver findings. In recognition of this, the work presented here incorporated both inductive and deductive approaches, using a deductive template to structure analysis, with explicit remit to highlight other issues that emerged inductively from the data, though the focus was on inductive analysis. The process is presented in detail in table 1. Researchers spent approximately 1 hour with each transcript or document, as stipulated by Hamilton in her description of the approach, noting key issues in a one-sheet structured 'summary template', with no formal coding. The data entered into the summary templates focused on the main issues in the data, rather than every single issue that surfaced. The RA summary template was made up of a number of sections describing participant and data collection details and deductive and inductive headings. At the end of the template, there were further sections to record key documents, observations, quotations and reflections relating to the data collection episode. The deductive aspects of the initial summary template were developed from the research questions: rationale for implementing the home birth service, programme design (structured according to logic model domains), facilitators and barriers to implementation and routinely gathered data about the service. This template was tested by both RA researchers as described in table 1. During this early testing process, it was deemed necessary to inductively develop a small number of additional subheadings for three of the template sections (rationale, barriers and facilitators) to help the researchers to organise the data. Although the use of more focused approaches has been highlighted to be of value when interpreting data for reporting in a health service context, the need to maintain a thorough and transparent process must go hand in hand with producing findings that are easily understood and relevant to stakeholders.[11] The summary template accompanies this paper (online supplementary file 1). Summarised data were explored with respect to the research objectives to produce a report summarising findings and recommendations.

### Secondary TA using the framework method

The secondary analysis was conducted by IL between June and September 2015, after the original RA was complete, with oversight and support from [researchers 5 and 6]; all three are experienced qualitative applied health researchers from outside of the original team. Typically, the purpose of secondary analysis is to explore new research questions,[21] but in this case, secondary analysis was performed using a different method to meet the same objectives as the primary analysis to compare the outputs of the two methods. The original team ([researchers 1, 2 and 3]) provided brief contextual details about the field, the organisations and participants involved and the background to the project. No further discussion occurred to avoid revealing RA findings to the TA team. The TA was informed by the original research objectives, using an inductive approach, and following the steps set out in the framework method, an approach to TA developed by Ritchie and Lewis[8 19]: familiarisation, coding, developing a framework, applying the framework, charting data into the framework, interpreting data and writing up. Table 1 summarises the process

### Notes on methods used

It is important to acknowledge that the creative and flexible nature of qualitative methods means that there is variation in the way different researchers undertake even established methods. While we refer to the methods with proper nouns, and summarise as 'TA' and 'RA' to provide clarity for the reader, it should not be assumed that these methods are 'fixed'. In addition, while we refer to the framework method analysis as 'TA', we acknowledge that the framework method is one of many approaches that fall within TA.[8] We provide a full description of our approach for transparency. It should also be noted that while both methods use matrices, the approaches are quite different in that TA involves the detailed, inductive coding of data, producing a detailed coding framework and more complex matrix that accounts more completely for the dataset. RA focuses on major issues identified in the data, no full coding occurs, and matrices are deductively constructed.

### The comparison

The comparative analysis was conducted between October 2015 and May 2016, comparing three aspects of the analyses: time taken, findings and recommendations. Each team recorded the time taken to perform every activity. Analytical activities were divided into two broad areas: 'data review and management' and 'data interpretation and report writing' as indicated in table 1. Summary

**Table 1** Description of the rapid analysis and thematic analysis

| | | Rapid analysis | Thematic analysis |
|---|---|---|---|
| The researchers | | Clinical background. | Not clinical. |
| | | Embedded in the field. | No prior exposure to field. |
| | | First time using RA. | Experienced in TA – no need to 'learn'. |
| | | Shared office, opportunity to interact. | No informal interaction/reflection. |
| | | Evenly shared workload. | IL Conducted most of the analysis. |
| | | Main focus of work. | Conducted alongside other core work. |
| | | Conducted over short period. | Project delivered over a longer period. |
| | | Focused on producing and 'crafting'* outputs for known stakeholders. | Less focused on the needs and expectations of stakeholders. |
| Epistemological position | | Theoretically interpretive, generic qualitative approach. | Theoretically interpretive, generic qualitative approach. |
| Data collection | | Interviews, focus groups conducted, documents gathered from participants by BT and CH facilitating focus group. | Provided with pregathered dataset. |
| Transcription | | Audio recordings transcribed by third party. Transcripts checked for accuracy by researcher. Participant name retained in transcript. | Pseudoanonymised transcripts and documents provided. |
| Timing | | During and following data collection process. | Analysis conducted postdata collection. |
| Ordering | | Interviews, then focus groups, then documents. Strategic participant data analysed first. | Documents, then interviews with strategic participants first, finally the focus groups. |
| 'Data management and review' stage | Early analysis | [researchers 1 and 2] dual analysed one interview transcript, inserting them into a 'summary template', organised according to the research objectives (see online supplementary file 1). They compared template content. The process and the template structure were reviewed and amended (some subheadings applied). A second transcript was subsequently dual analysed in the same way. The 'one hour per transcript' rule was not applied here, spending 1.5–2 hours on each. | An identical sample of three transcripts reflecting a range of job title and seniority were analysed independently by [researchers 4 and 5] and the themes that emerged discussed and finalised. These themes were arranged into analytical hierarchies, that is, consisting of the key themes and associated subthemes, and these formed the basis for the codification of the remaining data. |
| | Main analysis | Remaining data items allocated equally to [researchers 1 and 2], following the same process, limiting time to 1 hour maximum per data item (less for some less complex documents). Researcher entered information directly into a matrix, structured as the template, using individual templates duplicated work. | [Researcher 4] independently analysed the remainder of the transcripts and the resulting themes and subthemes were agreed with SG and formed the analytical hierarchy for the remaining data. |

Continued

**Table 1** Continued

| | | Rapid analysis | Thematic analysis |
|---|---|---|---|
| 'Interpretation' stage | Data interpretation | [Researchers 1 and 2] reviewed content in one another's matrices, and combined them. Data were allocated equally to [researchers 1 and 2] for interpretation and write up, organised according to the template, for example, facilitators to implementation. The 'barriers' section was more complex, and this was subdivided into themes, which were allocated to [researchers 1 or 2], for example, training, promotion and recruitment. A summary of findings and a set of recommendations were produced for each. Summaries were reorganised thematically. | IL undertook interpretation and write-up of the findings according to the thematic headings. For each theme and subtheme, an explanatory sentence was produced, and an exemplar quote or quotes was selected. These themes and subthemes were used to create a list of findings specific to each overarching theme. |
| | Final report writing | Summaries of findings and recommendations were combined and checked by [Researchers 1, 2 and 3] to eliminate duplication and reach consensus regarding interpretation, revisiting the primary data where necessary. | These findings were used to inform a final report, populating the template provided by BT. The report template included the following headings:<br>1. Participants and data (not written up in secondary analysis).<br>2. Timeline for development of service.<br>3. Service design (logic models developed for intended and actual service design).<br>4. Achievements.<br>5. Challenges<br>– Barriers to implementing the model as intended.<br>– Barriers to delivering specific service outcomes.<br>6. Service data<br>– Data being gathered.<br>– Responsibility for data collection/entry/analysis.<br>– What is going well in Home Birth Service (HBS) data capture and use.<br>– Data-related challenges.<br>7. Recommendations. |
| Researcher interaction | | [Researchers 1, 2 and 3] reflected and discussed the data and interpretation on a regular, iterative basis. | [Researchers 4 and 5] had several telephone and one face-to-face discussion. |

*'Crafting' refers to the writing and editing of findings and recommendations to present content and language deemed to be appropriate to the service stakeholders by the rapid analysis team.
HB, Home Birth Service.

**Table 2** Characteristics of the two research teams

| Rapid analysis researchers | Thematic analysis researchers |
|---|---|
| Clinical. | Lead researcher not clinical. |
| Embedded in field. | No prior exposure to the field. |
| BT collected the data. | Did not collect data. |
| Using rapid analysis for first time – developing new practice. | Experienced in thematic analysis – using existing skills. |
| Shared office. | No shared space. |
| Equal workload within team. | IL conducted majority of analysis. |
| Analysis main task at work. | Analysis conducted alongside other commitments. |
| Focused on producing outputs for known stakeholders. | Much less focused on the stakeholder team. |

statistics were produced using data from the resulting time sheets. Findings were defined as individual issues identified and included in a report. Recommendations were defined as suggested actions to improve or maintain the service. Each team then independently compared RA and TA findings, allocating a 'match', 'partial match' or 'mismatch' category. Both teams then met to discuss and reach consensus. Any mismatches were discussed, and perceived reasons were agreed and recorded and summary statistics was produced.

### Patient and public involvement (PPI)
This paper is a methodological exploration of two different means of qualitative analysis. There was no PPI involvement in establishing the criteria for comparison nor in facilitating the work. However, PPI was intrinsic

to the original programme from which the data were gleaned.[17]

## RESULTS
### The research teams
Table 2 presents the characteristics of the two research teams.

### Comparison of time
Table 3 illustrates the time taken at each stage of the process, for the 'management' and 'interpretation and report writing' stages defined earlier in table 1. The 4 hours of background discussions to provide IL with context were not counted in the total. The RA data review and management took around a third of the time of the TA (43 hours and 116.5 hours, respectively). The reverse was true of the report writing; RA was more than six times longer at 52 hours.

### Comparison of findings
The comparison of findings is presented in table 4. TA elicited marginally more findings than RA (153 vs 131). There were 107 matches. There are differences in reporting style and level of detail in the matched findings, with the example below highlighting how each team provided similar findings but with a varied degree of specific information. Both teams had examples where they provided more or less detail than the other on a specific topic, but the reporting style in the RA was consistently more 'polished', with findings more consistently framed in a way that would be more accessible to the intended audience (explored further in the discussion).

There are issues around communication with ambulances/paramedics. TA finding

**Table 3** Time taken to complete analysis using rapid analysis and thematic analysis

| | Rapid analysis team | | | | Thematic analysis team | | | |
|---|---|---|---|---|---|---|---|---|
| | | Time taken (hours) | | | | Time taken (hours) | | |
| | Activity | [R1] | [R2] | Total | Activity | [R4] | [R5] | Total |
| Primary data review and management | Review two transcripts and develop summary template | 6 | 5 | 11 | Review/code initial transcripts | 11 | 9.5 | 20.5 |
| | Refine template | 2 | 2 | 4 | Developing framework | 3 | 1 | 4 |
| | Complete summary template for remaining transcripts | 13 | 11 | 24 | Review/code remaining transcripts | 82 | | 82 |
| | Reviewing documents | 2 | 2 | 4 | Reviewing documents | 4 | | 4 |
| | Reviewing matrix | 2 | 3 | 5 | Final themes | 8 | | 8 |
| | Total | 25 | 23 | 48 | Total | 108 | 10.5 | 118.5 |
| Interpretation and report writing | Writing up findings | 16 | 16 | 32 | Writing up findings | 4 | | 4 |
| | Writing recommendations | 8 | 12 | 20 | Writing recommendations | 4 | | 4 |
| | Total | 24 | 28 | 52 | Total | 8 | 0 | 8 |
| Total | | | | 100 | | | | 126.5 |

**Table 4** Quantitative comparison of findings and recommendations elicited using rapid analysis and thematic analysis

| | | Rapid analysis | | Thematic analysis | | Total |
|---|---|---|---|---|---|---|
| Findings | Matched | 71 | 54% | 78 | 51% | 107 |
| | Partially matched | 28 | 21% | 37 | 24% | 43 |
| | No match found | 48 | 37% | 32 | 21% | 80 |
| | Appears in other team's recommendations (not findings) | 9 | 7% | 3 | 2% | 12 |
| | Total* | 131 | | 153 | | N/A |
| Recommendations | Match | 18 | 28% | 32 | 34% | 32 |
| | Partial match | 20 | 31% | 26 | 28% | 26 |
| | No match | 26 | 41% | 42 | 45% | 68 |
| | Total* | 64 | | 93 | | N/A |

*This does not reflect column total as findings/recommendations from one team frequently matched (fully or partially) two or more from the other team.

Some paramedics are unaware that the HBS exists and there have been delays of up to 30 min between the paramedics being informed of a BBA and this being cascaded down to midwives. RA finding

Findings from one method frequently matched two or more from the other: 71 RA and 78 TA findings delivered 107 matches. There were 43 partial matches, where findings identified similar, but not identical issues (28 RA, 37 TA, some matching more than once), for example:

There was a general consensus that useful meetings with a range of stakeholders were hard to arrange for a number of reasons including workload and shift pattern. TA finding

While support is strong in-principle, there is no formal process for strategic-level consultation and decision-making about the HBT within the provider Trust (outside of the Project Board). In addition, busy workloads make collaborative working challenging. RA finding

Eighty findings could not be matched: 46 or 37% of all RA findings and 34 (21%) of the TA findings. Exploration (see table 5) revealed that the most common reason for mismatches was that the other team simply did not interpret the relevant finding from the data. The TA team did not find 11%, and the RA team did not find 12% of the opposite team's findings. The next most common reason was that findings were specific or detailed, rather than key issues with broad relevance. The RA team also reported 15 positive findings (successes and achievements), which the TA team did not include in a report to the Service: the TA team reflected that they focused on constructive feedback about challenges and areas requiring improvement, rather than positive findings (explored further in the discussion). For example, the RA team reported 'The HBT MWs are generally supportive of the need for data collection and comply with this', and 'The Service has produced its first comprehensive data report for the Project Board (November 2014)'.

There were a small number of findings that emerged from interpretation of 'what was not in the data'. For example, the RA team reported that staff may not gain necessary qualifications for deployment, which was a risk to service resilience, connecting data on staff training with other data concerning service staffing requirements, rather than a direct report from research participants. The TA team did not identify this finding. The RA team's contextual knowledge meant that they perceived some TA findings to be incorrect. For example, a TA finding suggesting that regular meetings were helpful was rejected, as the RA team had been informed (outside of the formal data collection) that the meetings were not functioning as intended.

Finally, the RA team unconsciously suppressed two findings that were politically challenging: they agreed with these two findings from the TA team, which concerned relationships and performance of individuals connected to the Service (exact examples cannot be provided as they are of a sensitive nature). The RA team reflected that while they were aware of these issues, and also knew that the Service was aware of them, they did not write them up as findings in the report. This was not an actively documented, discussed decision-making process between the RA researchers; it was more implicit that they could not 'go there' in a report.

Some findings appeared to have no match, but cross-checking revealed that the finding aligned with the other team's recommendations (nine RA and three TA findings). For example, the RA found that staff had requested more emergency training, and the TA recommendations included provision of more emergency training.

In terms of topics, the mismatched findings covered a range of different issues for the service.

Both teams identified findings missed by the other team, which covered operational issues and leadership and management issues for the Service. The RA team identified findings that were not elicited by the TA team relating to strategic issues, promotion of the service and

Table 5  Suggested reason for mismatched findings and recommendations with examples

| | Suggested reason for other team not eliciting finding/recommendation | Rapid analysis (RA) | Thematic analysis (TA) | Total | Examples |
|---|---|---|---|---|---|
| Findings | Straightforward miss/error | 16 | 17 | 33 | 'There has been no Audit against NICE guidelines for contact (number of visits)'. TA |
| | Specific/detailed | 10 | 11 | 21 | 'Aromatherapy oils are expensive'. TA |
| | Positive finding not reported | 15 | 0 | 15 | 'Initial engagement visits to community teams by HBT members facilitated implementation'. RA |
| | Finding emerged from 'what is not in the data' – higher level interpretation | 5 | 1 | 6 | 'It is not known whether current MSW recruits will be successful in the 45 credit [training] module, and how Service needs will be met if they are not'. RA |
| | The embedded team's knowledge of the context meant they did not agree | 0 | 3 | 3 | Examples suppressed as sensitive. |
| | Suppressed as politically challenging | 0 | 2 | 2 | Examples suppressed as sensitive. |
| | Total | 46 | 34 | 80 | |
| Recommendations | Straightforward miss/error | 3 | 18 | 21 | 'Ensure that meetings are attended by as many of full and part-time workers as possible'. TA |
| | Recommendation emerged from 'what is not in the data' – higher level interpretation | 19 | 0 | 19 | "Consider whether services which fall outside of 'standard' maternity care should be routinely offered, for example, complementary therapies, hypnobirthing, pool provision, high frequency or duration of contact with women'. RA |
| | Embedded RA team's contextual knowledge meant that they did not agree with recommendation | 0 | 15 | 15 | 'Co-locate the HBS with other maternity services.' TA; the RA team knew that this was not possible at the participating hospital trust. |
| | Specific/detailed recommendations for a service dataset or audit | 0 | 9 | 9 | 'Frequency of texts between mother and midwives could be retrospectively collated to demonstrate improved accessibility'. TA |
| | Contextual knowledge was used to develop recommendation | 4 | 0 | 4 | 'Ensure that the HBT midwives are sufficiently familiar with Birth Centre/Delivery Suite facilities and protocols.' RA; the TA team assumed this would be the case already. |
| | Total | 26 | 42 | 68 | |

performance management (which were often positive findings about 'successes' not reported by the TA team).

## Comparison of recommendations

Quantitative comparison of recommendations is presented in table 4. The RA generated 64 recommendations, a third less than the TA. Eighteen of the RA recommendations matched to 32 of those from the TA, and the individual RA recommendations tended to bring together multiple issues and were 'crafted' in such a way as to provide a smaller, number of recommendations combining multiple points. For example, the RA recommendation below encompassed three separate TA recommendations:

> Working model: urgently consult regarding whether the model (shift pattern/on call volume/accrued time) is fit for purpose, and if it is, how MWs can be supported to avoid burnout. In addition, consider whether the Service can realistically attend BBAs within this model, and if not how this key objective for the Service can be achieved. RA recommendation

> Collect more precise data on which BBAs did or didn't need to attend. Then look at feasibility of HBS attending these women in the home. TA recommendation 1

> Determine the capacity of current staffing levels and shift patterns. TA recommendation 2

> Begin discussions with staff on preferences and flexibility in order to meet growing demand. TA recommendation 3

Some recommendations were more directly matched, for example:

> Require future recruits to have achieved the minimum numeracy/literacy standard. TA recommendation

> Be clear on the necessary baseline skills in numeracy and literacy that are required. RA recommendation

There were partial matches between 20 RA and 26 TA recommendations, for example.

> Ensure robust lines of communication are in place between Home Birth Service and community midwives. TA recommendation

> Routinely feed back to referring professionals to confirm booking with Home Birth Service, or transfer back to community midwives. RA recommendation

A further 26 (41%) of the RA recommendations and 42 (43%) of the TA recommendations had no match. Reasons are presented with examples in table 5.

The most common reason was that the other team did not identify a particular recommendation, RA did not find 18 (35%) and TA did not find 3 (12%). Four of these TA recommendations related to training of midwives, three were about organisation of meetings and the remainder had no common theme. The researchers determined that the midwife training recommendations

were important and had been an analytical blind spot for the RA team. Other mismatched recommendations were collectively determined to be of low importance by the researchers, except for the TA team's recommendation about projected milestones for the service.

The RA team made 19 recommendations based on 'what wasn't in the data', interpreting beyond the reported facts. The TA team made 15 recommendations, which the RA team did not support, as their contextual knowledge deemed them unworkable or inappropriate. Nine recommendations that were not found in the RA recommendations were from the TA team who made a detailed list of items for a future service dataset, while the RA team provided less specific recommendations regarding a future data set. Finally, four recommendations were determined to be made due to contextual knowledge of the RA researchers.

## DISCUSSION

### Principal findings

This study compared RA and TA methods applied to the same dataset to explore whether RA provides timely, accurate outputs for services. RA data management took around a third of the time of TA, but RA interpretation and write up took more than six times longer than TA. There was considerable overlap in the findings and recommendations between the two methods, with RA identifying marginally more findings than TA, and TA making marginally more recommendations than the RA. The comparison identified qualitative differences in the depth and detail of findings and recommendations in the two teams.

### Strengths and limitations of the study

#### Strengths and limitations in the RA and TA processes

The qualitative analysis processes followed by each team have been described in detail to enhance reproducibility and reliability. However, we acknowledge that work of this nature can never be reproducible due to the subjectivity of qualitative researchers and processes,[22] and the fact that research is a situated practice, where some aspects of the activity are beyond the control of the researcher.[23] In qualitative research, there is much debate regarding subjectivity, reflexivity and bias.[22 24] In the conduct of our work, we attempted to minimise 'bias' and described our methods in detail, though we have also retrospectively identified opportunities where others can mitigate this further in future work. The findings of research such as ours, which does reflect on and compare processes and findings in a systematic and detailed manner, can contribute to understanding the challenges faced by researchers.[25] The characteristics of the researchers are acknowledged and explored. Researchers were similar in that they were all experienced postdoctoral health services researchers, working in the same Institute for some time, arguably with similar cultures, though we acknowledge that the human, interpretive nature of qualitative research

means that standardisation or researchers within and between the teams is not possible. There were differences between the researchers (see table 2). These factors may have conferred variation in analysis and interpretation.

The RA team had greater contextual knowledge resulting from previous clinical exposure as health professionals and working closely with the service. This appeared to impart an underlying level of understanding that was critical to the findings and particularly recommendations. It is useful to think about the concept of research as a situated practice in the context of our work. This may be particularly relevant for researchers who are 'embedded' in some way within the service being researched. While such embeddedness can help to provide useful insights into the meaning and relevance of research findings, it is important to be aware that this may unconsciously influence data interpretation.[23] RA in a health service setting without this background knowledge may be inappropriate. Around a third of RA findings were not accounted for by the TA: RA generated a large number of additional findings, suggesting that closeness to the field and data may have conferred an advantage. It has been recommended previously that contextual information should be provided to secondary analysts to mitigate the lack of exposure to the field.[21] The intended comparison of methods and need to avoid conferring between teams meant that the TA only received brief information, rather than the rich, iterative contextual information that may be more typically provided within secondary analysis.

The RA was conducted for a specific group of stakeholders, and the interpretation, and crafting of findings and recommendations, was done with these individuals in mind. Though not conscious of this at the time of analysis, on reflection, we believe that this focus on a specific audience, in addition to [researchers 1 and 2]'s relationship and sense of reciprocity with the service, may have resulted in a more lengthy approach. We reflected that it also resulted in more focus on reporting positive findings, or 'good news' in the RA team, and suppressing negative findings that concerned individuals, which the RA researchers deemed inappropriate to report in an evaluation output that would be widely shared. This contrasts with the TA that was a 'desktop exercise', with no commitment to the research participants, which we feel made the process more straightforward, with less need for careful presentation of data. This provides a clear example of researchers navigating the 'politics of research', telling stories differently as a result of the different purpose and context of the research.[26]

A second factor in explaining the lengthy RA is that it is the first time that [researchers 1 and 2] have used RA. Adapting to a new method can take time, and discipline is required not to refer to more familiar, lengthier practices. However, the number and detail in the findings and recommendations in the RA (131 and 62, respectively) was similar to those in the TA (153 and 93). For qualitative researchers trained in TA, it may be difficult to wholly adopt the brevity required of RA.

The TA was predominantly conducted by one researcher IL, providing fewer opportunities for reflection in the TA development. The RA team also had the opportunity for ongoing regular reflection due to shared office space, which may have enhanced but also lengthened the process.

Our approach to this work was pragmatic, based on available researcher capacity, and there was variation in researcher characteristics, in their programmes of existing work and embeddedness in the field for this study, which may have impacted on the outputs from the work. In future comparisons, involving some or all of both teams in data collection would provide equality in exposure and embeddedness, and increasing similarity in researcher characteristics could provide further parity. The workload and capacity issues are more problematic. The time taken to undertake analysis varies from project to project, based on the available time, deadlines, funding and competing priorities. Generally, there is always scope for extended analysis of data to explore it further, and researchers must make pragmatic decisions about when analysis for a specific project is 'finished'. It is likely that there is variation between decisions to cease analysis between research teams, particularly in our comparison, where the analysis was a 'desk top exercise' for the TA team and a 'real' project with stakeholders expecting outputs from the RA team, meaning the latter may be more inclined to spend longer on the project. To mitigate this, increased parity across the RA and TA researchers could be achieved by using two equal-sized teams, with equal division of labour, and explicit allocation of capacity to the project. However, it is still impossible to standardise decisions regarding what constitutes 'enough' work on a dataset.

### Strengths and limitations in the comparison process

This paper has provided an opportunity to explore and reflect on approaches to comparing qualitative methods. The limited evidence base necessitated the development of the comparison methodology. The study team regularly met to review the process, emerging findings and interpretation to enhance the rigour of the exercise. A mixed methods approach was undertaken in order to explore RA, which allows for a broader exploration of a phenomenon (the analytical process) than quantitative or qualitative methods alone.[27–29] However, the qualitative aspect was restricted to evaluation of the alignment content outputs of the research and description of the researcher characteristics and activity diaries by the researchers themselves. Future comparisons of methods could be strengthened with the addition of independent qualitative evaluation of the research processes and outputs. A limitation of the quantitative approach to comparing outputs from qualitative work is that it reduces findings and recommendations, directly comparing individual outputs that display different levels of depth and detail. It is important to highlight that 'more' does not necessarily equal 'better' in qualitative research outputs.

An important consideration when undertaking comparison of methods is the variation in processes between individual researchers. For example, while TA using the framework method follows an established process described in the literature, it is acknowledged that the complex nature of qualitative analysis, and the role of the researcher in the process, means that there will always be variation between researchers in the exact physical and cognitive processes involved. It is therefore not possible to 'standardise' between researchers, within or between the two methods being compared. While we perceive comparisons of this nature to be worthwhile in order to develop and understand the applications of qualitative methods, they must include detailed description of and reflection on the processes and researchers.

The complexity of the process only became clear once the researchers began to compare the data. Differences in style and the degree of 'polishing' of the content and language with the RA team 'crafting' findings and recommendations deemed sensitive and appropriate to be shared with stakeholders, and the resulting impact on time taken was not apparent until analyses were complete and outputs shared. In addition, devising an approach to categorising and reporting mismatched findings and recommendations took time and was not as intuitive.

A further limitation is the fact that the comparison was conducted by the researchers themselves due to pragmatic resource constraints. While we acknowledged this and aimed to maintain objectivity, there is clearly a risk of bias in interpretation, and future projects should consider involving an independent, blinded third party to conduct the comparison.

An unexpected outcome of this study is that it has encouraged us to reflect deeply on our own research practice, resulting in a better understanding of our methods and role. Future comparisons may benefit from independent exploration of the researchers' individual processes alongside the 'outcomes' of time, findings and recommendations. It is clear that there are a number of barriers that may constrain the research process in a service evaluation of the type we conducted. Greater reciprocal appreciation that these exist, and what they are, may help to facilitate discussions where there are unexpected or unpalatable research findings.[30]

The initial intention was to involve participants in reviewing the importance of mismatched findings and recommendations. This was not practicable due to the unexpected length of time taken to complete the comparison, and the need for service stakeholders to determine whether mismatches would have been helpful many months in the past.

It is important to note that all researchers in this study were experienced in qualitative health research using TA, and as such this study does not explore RA and TA for novice researchers.

### Possible explanations for the differences in time taken to conduct analysis

The time taken in the RA was much shorter at the data review and management stage, equating to around 2 weeks less whole time equivalent (WTE) researcher time. This suggests that managing data in this way within a short timeframe is possible. However, the interpretation and reporting phase was much longer with RA (6.5 days vs 1 day in TA). A number of factors may have contributed. Time saved in coding and data management may result in more time being required at the interpretation stage in RA. This needs further exploration; RA only took three WTE researcher days less that TA, which may be of little benefit to academic or health service stakeholders. There are further possible explanations: the researchers' relationship with the service, the purpose of the research, the capacity of researchers and the fact that the RA team were learning a new skill. This is explored earlier in the strengths and limitations section.

### Possible explanations for the difference in findings

The RA findings accounted for 78 of the 153, or 79% of the findings delivered by the TA. This considerable overlap indicates that TA, which codes all data, did not produce many additional findings. This is consistent with others' findings comparing themes generated from different analytical approaches.

The most common reason for mismatches in findings was that the researchers had not identified the issue in error. In the RA, patterns and findings may have been missed as a result of the more deductive approach taken and the reduced time spent with primary data. However, there was a 'did-not-find rate' of around 1 in 10 for both methods, suggesting that this was not the case. The mismatches suggest that qualitative researchers will never elicit perfectly overlapping findings, regardless of method.

A number of mismatches were accounted for by unconscious suppression of challenging findings, higher level interpretation and differences in contextual knowledge leading to the rejection of findings. These explanations were more prevalent in the RA team, suggesting that embeddedness influences these processes. Between a quarter (RA) and a third (TA) of the mismatched findings were somewhat detailed, highlighting differences in natural reporting style, interpretation and prioritisation of what was meaningful. Again, this may arise between different researchers, regardless of method. Mays and Pope[31] relate how observations are '*limited by definition to the perceptions and introspection of the investigator*', and variations in perception and introspection are inevitable between different individuals. There are different views regarding whether qualitative findings should be reproducible,[32] but we take the stance that subjectivity and individual variation make this impossible. This has been a useful exercise in reflexivity, demonstrating how experiences and unconscious processes impact on findings.

The TA team did not report positive findings, accounting for a further portion of the mismatch: this was attributed to differences in interpretation of the project scope, rather than analytical processes delivering different results. Also, the TA team were aware that they would not be presenting findings to providers, meaning that they felt more able to be critical and candid.

### Possible explanations for the difference in recommendations

The recommendations also demonstrated overlap, with around three out of five being accounted for by both teams. However, RA did not pick up a third of the TA recommendations. We perceive that the majority reflected relevant but non-essential detail, and the 'make or break' recommendations that were key to the sustainability of the service were not missed, though we acknowledge that this is a subjective judgement. Arguably, the most important recommendation missed related to training midwives in administrative and management skills. This detail is consistent with the TA process, where the data were explored in more depth, leading to more precise recommendations. However, this pattern was not observed in the findings. A possible explanation is that the RA team, with the eventual audience in mind, were more conservative in the number and detail of recommendations. Over half of RA recommendations that the TA did not find were accounted for by higher level interpretation and contextual knowledge, and just under half of the TA mismatched recommendations were deemed inappropriate by the RA team due to contextual knowledge, suggesting that embeddedness in the field confers advantages, separate from the method used.

### CONCLUSION

We found that RA was appropriate and delivered valid findings and recommendations, with reassuring but not complete overlap. Mismatches appeared to relate to minor or detailed issues. RA enabled considerable time savings in data management but may not be as rapid as assumed. Further work is needed, addressing the limitations identified to establish how much time experienced RA researchers can save, whether differences in outputs are due to the analytical method or other influences and whether these are relevant and of practical benefit for stakeholders and to services. Researcher characteristics, conduct and roles are key, and our impression is that RA requires the researchers to be embedded in the field.

We do not advocate RA for granular exploration of complex questions, for example, individuals' experience of phenomena. It could be used to rapidly identify issues for further, in-depth qualitative exploration. RA represents one of many tools of the qualitative researcher's trade, with particular potential for use in applied health research, when timely reporting is needed. We advocate further work to identify the practical application and use of different rapid approaches in practice.

**Acknowledgements** We are grateful to Louise Bentham, who conducted thematic analysis of three of transcripts and reviewed findings with IL. We extend our thanks to the hospital trust who initiated the primary research project and granted permission for the work and to the staff who gave up their time and shared their perspectives so generously. We are grateful to the Scientific Advisory Committee of the West Midlands Collaboration for Leadership in Applied Health Research and Care, who provided feedback on initial findings and interpretation. CH acknowledges support from the NIHR Oxford cognitive health Clinical Research Facility.

**Contributors** The original idea for the project was conceived by SK. The study was designed by BT with intellectual input from all authors. Primary data collection was conducted by BT. Rapid analysis of data was conducted by BT and CH, with input from SK. Thematic analysis of data was conducted by IL with input from SG. Comparison of time data was conducted by BT and checked by CH and IL. Initial comparison of findings and recommendations was conducted by BT, CH, IL and SK, and all authors reviewed outputs from the comparison. The preliminary draft of the paper was written by BT. This was critically reviewed by CH, IL, SG and SK for important intellectual content and subsequent revisions to the paper were undertaken by BT as a result. Final approval of the version of the paper to be published was granted by BT, CH, IL, SG and SK, who all also agreed to be accountable for all aspects of the work in ensuring that questions relating to the accuracy or integrity of any part of the work are appropriately investigated and resolved.

**Funding** This work was supported by the National Institute for Health Research (NIHR) through the West Midlands Collaborations for Leadership in Applied Health Research and Care (CLAHRC-WM) programme.

**Disclaimer** The work was developed independently by the authors, and the views expressed are those of the authors and not necessarily those of the NHS, the NIHR or the Department of Health.

**Competing interests** None declared.

**Patient consent** Not required.

**Ethics approval** The primary service review and secondary analysis were reviewed by the University of Birmingham Ethics Committee, ref ERN_15–0127S.

**Provenance and peer review** Not commissioned; externally peer reviewed.

**Data sharing statement** Data are securely stored at the University of Birmingham in line with our information governance and data protection policies. Due to the confidential nature of our qualitative data, which may identify individuals even following anonymisation, we have not made the data publicly available, in line with our research permissions and consent.

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
