## [Reviewer comments · BMJ Open]

This paper was submitted to a another journal from BMJ but declined for publication following peer review. The authors addressed the reviewers' comments and submitted the revised paper to BMJ Open. The paper was subsequently accepted for publication at BMJ Open.

(This paper received three reviews from its previous journal but only two reviewers agreed to published their review.)

ARTICLE DETAILS

TITLE (PROVISIONAL)	Can rapid approaches to qualitative analysis deliver timely, valid findings to clinical leaders? A mixed methods study comparing rapid and thematic analysis
AUTHORS	Taylor, Beck; Henshall, Catherine; Litchfield, Ian; Kenyon, Sara; Greenfield, Sheila;

VERSION 1 – REVIEW

REVIEWER	Sara Garfield Imperial College London NHS Trust
REVIEW RETURNED	11-Oct-2017

GENERAL COMMENTS	This is an extremely interesting paper and I enjoyed reviewing it. The paper is comprehensive and reflective but could be tighter and clearer in places. I have given some examples below but I would suggest a thorough proof read: 1. Table 3 seems to be missing units of time2. Are there some words missing on Line 317? i.e should it read 'which were not found in the ra recommendations.''3. Line 337. I am not sure how table 5 relates to the sentence preceding its reference.4. There are two very similar titles in discussion, 'strengths and limitations of the study' and 'strengths and limitations of the comparison. I wasn't sure of the difference between the 2 sections and was wondering if it would be better to combine them.5. The conclusion in the main text would be strengthened by being briefer and more snappy.6. It may be helpful to the reader to define what is meant by 'data management 'and 'data interpretation' and the difference between these 2 processes early in the paper.
--

REVIEWER	Kristina Wolff The Dartmouth Institute for Health Policy & Clinical Research, Dartmouth College, Hanover NH USA
REVIEW RETURNED	09-Nov-2017

GENERAL COMMENTS	I reviewed this article with hope that rapid approaches could be a useful addition to health services research. However, I have some
--

	concerns regarding the methods used in conducting this study. The most serious concern to me is lack of attention paid to places where bias may be introduced into the organization of the research teams, data collection and analysis. This is easily found in Table 1 where it is noted that the RA team “Focused on producing and ‘crafting’ outputs for known stakeholders”. How does this not introduce bias into the analysis in favor of providing a positive outcome as well as ignoring findings that don’t match the chosen outputs? Whereas a thematic analysis looks for patterns that emerge and if done well, will minimize bias. If one is looking at qualitative data to look for things that impact outcomes in a healthcare setting, the goal is often to find things that aren’t discovered using traditional quantitative approaches. It is not clear to me how the RA approach would be preferable as explained in this study. For the main analysis it is unclear to me why the RA team was limited to a one-hour period to analyze a data item when there was no time limit for the TA team. Doesn’t this introduce bias into the amount of time taken for the analysis and therefore the RA team would automatically go faster because of this limit? How then is this a significant finding as reported on pg. 10 where the authors’ note that the RA took one third the time. This was influenced by the pre-set limits of the researchers. Perhaps it took the RA team longer to report because they didn’t get the same depth of information as the RA team. This is clear when the researchers tried to match the findings. To me this makes using RA as described in this manuscript as being a preferable qualitative approach. I think the interpretation section is unclear as they way in which the RA team organized their content (via matrices) may be similar to how someone using TA operates as well. I may be misunderstanding what they did, but my understanding is they created a false construct based on an assumption that TA is done the same way by qualitative researchers, but I know some TA who organize their material in similar ways to how the RA team did via themes / matrices, etc., it is unclear to me how this step is dramatically different then how a TA approach may organize their material. There needs to be more information here to have a meaningful comparison. I think this step is another area where bias would be introduced on the RA side. Due to these uncertainties and areas where I think bias was introduced in their methods, I am unconvinced that their findings are “valid” and “reliable” or that they’ve shown that the speed of the RA is worth sacrificing important information that is found using a TA approach. Therefore I cannot recommend this study for publication.
--	--

REVIEWER	Joanna Reynolds London School of Hygiene & Tropical Medicine, UK
REVIEW RETURNED	06-Dec-2017

GENERAL COMMENTS	I enjoyed reading this paper and think it holds potential for useful methodological learning in applied health research. However, I feel some revisions are required to draw out what I see as the more interesting and more important findings from the study: the dynamics and politics of conducting qualitative analysis in different ways and different contexts, rather than simple comparisons of the number and comparability of findings identified by two approaches. I've made some more major and minor recommendations in the attached document.
--

VERSION 1 – AUTHOR RESPONSE

Reviewer	Comment	Reviewer comment	Author response	Author action
1	1	Table 3 seems to be missing units of time	Thank you for highlighting this omission	Units have been added to Table 3 (hours)
1	2	Are there some words missing on Line 317? i.e should it read 'which were not found in the ra recommendations.'	This would be a useful clarification, thank you	Additional words added to text, now line 412.
1	3	Line 337. I am not sure how table 5 relates to the sentence preceding its reference.	Thank you, this is an error and should read 'Table 2'	Amended text to 'Table 2'
1	4	There are two very similar titles in discussion, 'strengths and limitations of the study' and 'strengths and limitations of the comparison. I wasn't sure of the difference between the 2 sections and was wondering if it would be better to combine them.	In a traditional qualitative paper we would present the strengths and limitations of the approach to qualitative analysis ('the study'). In this paper we then go on to undertake a separate comparison of methods process ('the comparison'), which has its own strengths and limitations. Our preference is to maintain the separation between 'the study' and 'the comparison' as two distinct processes, to increase the clarity of the paper.	No amendment
1	5	The conclusion in the main text would be strengthened by being briefer and more snappy.	Thank you for this suggestion.	The conclusion has been shortened to make it more succinct.
1	6	It may be helpful to the reader to define what is meant by 'data management 'and 'data interpretation' and the difference between these 2 processes early in the paper.	Thank you for this recommendation, we have amended Table 1 to more clearly reflect the two stages and highlighted this in the text.	Table 1 amended, explanatory text added (line 258-259).
2	1.	The most serious concern to me is lack of attention paid to places where bias may be introduced into the organization of the research teams, data collection and	In Table 1 we summarise the differences between the teams which might impact on the findings. We are grateful for the recommendation for	 We have explored the impact of researchers, embeddedness and research as a situated practice in more detail (line 356-361, 395-411).

		analysis. This is easily found in Table 1 where it is noted that the RA team “Focused on producing and ‘crafting’ outputs for known stakeholders”. How does this not introduce bias into the analysis in favor of providing a positive outcome as well as ignoring findings that don’t match the chosen outputs? Whereas a thematic analysis looks for patterns that emerge and if done well, will minimize bias. If one is looking at qualitative data to look for things that impact outcomes in a healthcare setting, the goal is often to find things that aren’t discovered using traditional quantitative approaches. It is not clear to me how the RA approach would be preferable as explained in this study.	further reflection on where bias may be introduced, and on the nature of ‘crafting’. It is important that we acknowledge the more deductive approach of RA and how this may increase the chance of ‘missing’ findings (distinct from the chance of missing findings due to the speed of analysis). The RA approach was hypothesised to be preferable as it would deliver findings quickly, and capture the most important findings that can then be translated into practice impact, and we sought to explore whether this was the case, and to examine the differences in findings delivered by RA compared to TA.	 • We have added some text and a reference to the Primary Rapid analysis section within the Methods to stress that whilst outputs must of course be relevant to stakeholders, maintenance of rigour and transparency in the research process is of paramount importance (line 215-218). • We have added text to clarify the nature of ‘crafting’ (239-240) We reflect on this process in lines 371-382. • We have acknowledged the more deductive approach and possible impact more clearly (line 195-201, 486-488). • We have identified the potential impact of the rapid approach, rationale for balancing speed with depth/completeness (line 195-201).
2	2	For the main analysis it is unclear to me why the RA team was limited to a one-hour period to analyze a data item when there was no time limit for the TA team. Doesn’t this introduce bias into the amount of time taken for the analysis and therefore the RA team would automatically go faster because of this limit? How then is this a significant finding as reported on pg. 10 where the authors’ note that the RA took one third the time. This was influenced by the pre-set limits of the researchers. Perhaps it took the RA team longer to report because they didn’t get the same depth of information as the RA (TA?) team. This is clear when the researchers tried to match the findings. To me this makes using RA (TA or RA?) as described in this manuscript as being a	The time limit was a key aspect of the RA, as described by Alison Hamilton in her description of the overall approach, and is a strategy to speed up the process of qualitative analysis, we apologise that this was not completely clear and have amended the text. Our comparative analysis is designed to explore whether the process is, in reality, faster, and also describe the outputs of the two methods: we expected there to be a difference for the reasons highlighted by the reviewer. The early stage of RA carried out according to Hamilton is, by definition, shorter, and it is important that we describe the differences between the approaches accurately by providing detail	We have clarified that the one hour is stipulated by Hamilton (line 203).

		preferable qualitative approach.	about the processes, time data, followed by reflection upon its impact, which follows in our description of the time taken to undertake the later stage of RA. We reflect why we may have taken longer to report beginning on line 409, “The time taken in the RA was much shorter at the data review and management stage, equating to around two weeks less whole time equivalent (WTE) researcher time. This suggests that managing data in this way within a short timeframe is possible. However, the interpretation and reporting phase was much longer with RA (6.5 days versus one day in TA). A number of factors may have contributed. Time saved in coding and data management may result in more time being required at the interpretation stage in RA. This needs further exploration, RA only took three WTE researcher days less than TA, which may be of little benefit to academic or health service stakeholders.”	
2	3	I think the interpretation section is unclear as they way in which the RA team organized their content (via matrices) may be similar to how someone using TA operates as well. I may be misunderstanding what they did, but my understanding is they created a false construct based on an assumption that TA is done the same way by qualitative researchers, but I know some TA who organize their material in similar ways to how the RA team did via themes / matrices, etc., it is unclear to me how this step is dramatically different than	Thank you for highlighting this important point. We recognise that TA is conducted differently by different researchers, but have not acknowledged this clearly in the text, and have amended the text to explore the variation even within methods. We have also added a further explanation of the difference in use of matrices between the two methods.	Text added in methods and limitations sections (line 250-254, 29-437).

		how a TA approach may organize their material. There needs to be more information here to have a meaningful comparison. I think this step is another area where bias would be introduced on the RA side.		
2	4	Due to these uncertainties and areas where I think bias was introduced in their methods, I am unconvinced that their findings are “valid” and “reliable” or that they’ve shown that the speed of the RA is worth sacrificing important information that is found using a TA approach. Therefore I cannot recommend this study for publication.	We hope that our comments and related amendments to the paper address the uncertainties identified. Our methods were pragmatic, and there were key differences between the approaches, and we have attempted to faithfully describe these. We recognise that this is not a ‘perfect’ study, but makes an important contribution in that it explores a previously underresearched area, and sets out clearly our methods for comparison, and reflects on how comparison of qualitative methods might be undertaken in the future. We highlight that RA is not a replacement for TA in all but is likely to be a useful tool which will be more appropriate in some circumstances than others. Our findings are nuanced, and we do not wish to make explicit claims regarding the speed benefits versus information sacrifices. We have not identified a large time saving by using RA as described, but we posit that this may be possible if the approach was repeated, and explore the reasons why. While there were differences in the RA and TA findings, we do not consider the TA to have revealed more ‘important’ findings than the RA, and hope	No action, but we would welcome further feedback

			that this is communicated in the text. We suggest that while RA shows potential, further exploration is required to understand its validity, reliability and application in health services research.	
3	1maj	I think you should be much clearer from the outset of the paper about your epistemological position in relation to approaching qualitative analysis, to help contextualise your approach to this topic, and to bring clarity to your assumptions about what different qualitative analysis approaches can and cannot do. You hint at your position towards the end of the Discussion where you say that “subjectivity and individual variation” make reproducibility of qualitative analysis impossible (which I agree with); but it would strengthen the paper to make your position clear from the outset. In addition, you should then the language of the paper to be sure it doesn’t contradict the position you set out; for example, in the Strengths and Limitations section you state that you’ve described your qualitative analysis in sufficient detail to “enhance reproducibility”.	Thank you for highlighting this important point. We will amend the text accordingly.	We have added our epistemological stance to the methods and Table 1. We have emphasised the caveats around reproducibility and standardisation within qualitative research in the discussion (line 337-352).
3	2maj	I have a slight issue with the way you present the analytical approaches as fixed (for example presenting them as proper nouns with capital letters), and as if there is one, agreed way to do them. This	Thank you for highlighting this, in our attempt to present the process in an uncomplicated way, on reflection we have omitted this important point. Another reviewer has also identified the need to be clear	We have provided further detail on the approach taken here, and acknowledge the creativity and flexibility in qualitative research (line 243-254).

		might be more the case for rapid analysis, but thematic analysis is not one single thing. You need to consider the different ways in which thematic analysis is framed, applied, described across methodological literatures, and define your own take on what it is and how it is done. I think it's risky to imply that qualitative analytical approaches are as regimented and neat as, say, statistical approaches – their value often comes in the creativity and flexibility with which they can be applied.	that thematic analysis is not a single thing.	
3	3maj	There is space (and need) for more engagement with methodological literature, which is fairly weak in the paper at the moment. You could refer to other 'rapid' approaches to qualitative research that have been developed recently, for similar reasons, for example the literature on rapid ethnographic assessment. In the Discussion, you need to situate your findings in relation to more literature for example on conducting mixed methods and multidisciplinary research, the 'politics' of doing research, and research as a situated practice, rather than a 'clean' rational process. You draw out some interesting points in the Discussion that could contribute to these literatures, so I recommend engaging with them.	Thank you for providing advice and direction to enhance the connection with existing work.	We have enhanced this section to engage further with the methodological literature (line 103-118). We have added reference to mixed methods literature, politics of research, and research as a situated practice (lines 418-420, 337-346, respectively). We have added some text and a reference to the Strengths and Limitations section which draws on work on ethical issues in qualitative research (lines 337-346), which suggests that 'misinformation and misinterpretation' are important things to bear in mind and that 'guidelines are needed'. We suggest that the findings of research such as ours which does reflect on and compare processes and findings in a systematic and detailed manner can contribute to understanding the challenges faced by researchers. We have been explicit in our paper as to the

				researchers' different positions visa vis the research environment and the influence this can potentially have on data interpretation . We have added some text regarding research as a situated practice in the Strengths and Limitations section to pick up this point and included a supporting reference (375-382).
3	4maj	I feel you could push your results further. For me, the counts of matches and mismatches is not particularly interesting without understanding more about the types of topic / issue that did or didn't match, more about the different levels of depth, framing and detail of these matching / mismatching results and recommendations. You start to explore the differences in level of depth, but I think should go further with this (and reduce the focus on number of matches). In the Discussion, the emphasis on number of results identified in each approach suggests an assumption in that 'more' findings from qualitative analysis equals 'better' analysis – I think you should unpack this assumption a bit more and justify it, especially given that there were some differences in the level of detail.	Thank you for identifying how we can develop our findings to be more interesting and useful to readers and highlight key issues with comparing qualitative analysis outputs. In the original paper we provided examples of data extracts which constituted 'match' or 'mismatch' in the reviewed paper, commented on the different ways in which finding matched, and explored the mismatched findings in Table 5. We have added further detail regarding the depth, framing and detail.	We have added further detail regarding the topics, in particular the mismatched findings line 346-351 (detail is already provided on the recommendations in line 385-390). Further detail on the detail and framing of findings is provided in lines 286-290, and on recommendations from line 365-379. We have also added additional reflection on this topic in the discussion (lines 424-427).
3	5maj	You should also take more space to explore the political dimensions of the analytical process that your study helps to illustrate. I think a really interesting point which you	We agree that further detail and reflection would strengthen the manuscript, and have expanded on our discussion of	Discussion of 'not useful' extended and examples provided in the findings: Lines 315-321,

	gloss over briefly is that the RA team “reported positive findings which the TA team did not deem useful” (line 256) – elaborate on this, perhaps with some examples. What were these findings and why did the TA team deem them ‘not useful’ (and not useful for what – for improving the programme, or for meeting the objectives of the evaluation?). This is the kind of issue that your paper should engage with more; it’s of far more interest and value from a methodological perspective than simple counts of matches / mismatches. You also identify some of the political issues underpinning analysis and comparing the two approaches, eg ‘incorrect’ interpretations due to a lack of contextual knowledge on the TA part, and the suggestion that the RA team ‘unconsciously suppressed’ ‘politically challenging’ findings (define what you mean by ‘unconsciously suppressed’). Again, I think it would be much more valuable for you to explore these tensions in detail, as they tell us a lot about the process of conducting qualitative research as part of mixed-methods evaluation, including aligning qualitative interpretations with knowledge from other areas of the evaluation, and the politics of presenting and representing knowledge and recommendations.	our interpretation of the comparison of findings.	We have elaborated on the ‘unconsciously suppressed’ definition and our reflection regarding this (lines 332-338, 375-378). The sensitive political nature of this aspect makes it difficult to provide concrete examples as it concerns individuals rather than broader service issues. See earlier response to comment on the politics of research for explanation of our action in this area.
--	--	--	--

3	6maj	Relating to this is the issue of time taken for each approach. Although I think it's useful to reflect on the time taken for each activity, there's still another issue that often analysis in research studies takes as much time as can be pragmatically allocated to it; there's always more that can be done, more avenues to explore, more issues to unpack, but we often have to make very pragmatic decisions about what is 'enough' and when to stop based on broader things such as overall study timelines, deadlines for funders, competing priorities etc. It would be helpful for you to reflect on this as part of your interpretation of the findings.	This is a very salient point, reflecting the realities of qualitative research practice. We have amended the text accordingly.	New text added lines 395-411.
3	7maj	In the Discussion you make the important point that it would be beneficial to explore with stakeholders the significance of mismatched findings / recommendations. Here you could refer to different bodies of literature on what kinds of 'evidence' and knowledge are considered useful by practitioners / policy makers, as your work could contribute to this field. What we as researchers think is important in terms of level of detail etc of reporting research findings is often at odds with what other stakeholders want / need.	Thank you, this would be a useful addition to the paper and we have amended the text accordingly.	We have added text to the discussion and a reference to the literature to make the point that greater reciprocal appreciation of the different barriers that exist on each side may help to facilitate discussions where there are unexpected or unpalatable research findings (lines 455-458).
3	8maj	I think the Discussion needs some restructuring. Parts of it feel like an extension / repetition of the Results section. You should avoid	Thank you, we have restructured the findings.	Findings summary reduced (lines 323-330).

		repeating too many of the findings (eg numbers of matches / mismatches) and go further to interpret the meaning of your results in relation to existing methodological / theoretical literature. Also, under Strengths and Limitations you describe some of the strengths and limitations of the two analytical approaches – these reflections are your study findings rather than S&Ls of the study itself.	Please see response to comment 3 regarding methodological literature. We have reflected at length on the comment regarding moving reflections to the strengths and limitations section, which has only been identified by reviewer 2. We would like to explain our rationale. We have approached the comparison write up in a similar way to a quantitative intervention study, where a paper would usually explore the limitations of the procedure for the two interventions under comparison in the strengths and limitations section, alongside the limitations of the comparison methods. We would therefore prefer to retain our reflections about the TA and RA approaches in the strengths and limitations section, but would be grateful for further advice on this.	
3	1min	1. Abstract: you should state what the data were that were analysed through the two approaches (ie a combination of interview and focus group transcripts)	Thank you for this recommendation, we have amended the text accordingly.	Amended as advised line 57.
3	2min	2. P4, line 95 – I’m not sure what you mean by ‘continued user experience’ – could you explain or revise the wording.	We agree that the language here was not clear, and have amended the text.	Amended as advised line 95.
3	3min	Background, lines 93-96: there are additional uses of qualitative research in applied, mixed methods health research: to explore in more depth questions or issues identified through quantitative work; to problematise or ‘unpack’	Thank you for the recommendation: while we had not provided an exhaustive list of applications, we welcome these additions and have amended the text accordingly.	New text added line 96-98.

		issues or topics taken for granted. Consider adding these in.		
3	4min	Line 103: I think you need to contextualise further your claim that interpretation of 'traditional qualitative approaches' takes a long time; presumably you mean in comparison with quantitative approaches (although some surveys etc can take a long time to administer, clean and analyse). Also, be clear that you're focusing in this paper on qualitative analysis approaches rather than data collection processes.	We agree that clarification is required and have amended the text.	New text added lines 103 and 116-118.
3	5min	5. Be careful with terminology eg p4 lines 106 to 108: I think you mean 'data management' rather than 'data collection' when you refer to using untranscribed audio recordings (these are representations of data already collected); and 'analysing data' instead of 'managing data' when you talk of summarising as opposed to formal coding.	We apologise for this error, and we will amend the text accordingly.	Table 1 'Data management and review' stage Text amended lines 108-114.
3	6min	6. Lines 115 – 116; could you clarify what you mean by 'focused on content analysis rather than interpretation' – do you mean this study only conducted content analysis and didn't compare this analytical approach with any other, or that they compared content analysis with a more interpretive analytical approach? If the latter, it would be helpful to understand what the findings of this study were.	Thank you for highlighting the lack of clarity here. This sentence introduces the scant literature that we have identified which describes comparisons of different methods. The studies 'predominantly...focused on content analysis rather than interpretation' in that they number and content of codes, rather than exploring the detail and interpretation of the resulting findings, and we have refined the text for clarity.	'Content analysis' replaced with 'number and content of codes', line 123.

3	7min	It would be helpful to give a (very) brief description of what the home birth service programme was, and how the interviews and focus groups fitted into the rest of the original evaluation study – it's not very clear to me at the moment.	We agree and have added additional text as indicated.	Further detail added lines 151-154, 165-166, 189-190.
3	8min	Clarify: was the focus group only conducted with midwives, or other staff members as well? How many SSIs were conducted, how long did they take etc? Lines 162 – 163 imply there was more than one focus group, but perhaps you meant to say that 'Interviews and the focus group each lasted approximately one hour...)?	Thank you, we have amended the text to clarify this	Text amended as advised lines 168-176.
3	9min	9. How did you devise the structured summary template used for the RA approach? What informed this? How did you adopt both deductive and inductive approaches – please give more detail. Given that you describe other ways of doing rapid analysis in the background (eg using audio files rather than full transcripts), I think you need to give more detail (in the main body of the paper) about your choices regarding what you actually did for the RA method. You cite Hamilton's approach, but don't give more detail in the main part of the paper. I think it's really helpful to understand what decisions were made and why for the RA approach.	We agree that further explanation strengthens the paper, and we have added more detail about our approach to RA.	Text added lines 105-221.

3	10min	10. Similarly there needs to be more detail regarding the choices made within the TA approach: it's not clear if only a sample of 3 transcripts were analysed or, the rest of the data set was analysed after devising the 'analytical hierarchies' from the early coding. Could you explain what you mean by 'analytical hierarchies' and how these informed the framework used for the TA analysis?	Thank you for identifying this gap in the description.	We have added additional description regarding the TA process in the TA column of Table 1, in the 'early analysis' and 'main analysis' sections of the table.
3	11min	Unclear in Table 1 what the template for report writing provided to the researchers using TA was and how it was structured – this might have influenced the compatibility of the two approaches.	Again we are grateful to reviewer 2 for pointing this out.	We have added the headings in the report writing template into Table 1 in the 'final report writing' row and the 'TA' column.
3	12min	P11 line 253: what do you mean by 'confirmed by returning to the original data'? Surely each finding was 'present' in the original data, it's just that some were interpreted by the RA researchers and others were interpreted by the TA researchers. I think you should be careful with language again here; thematic analysis is an interpretive approach, therefore it's not really appropriate to suggest that some findings were and were not 'found', as if they exist in the data independently of the researchers' interpretations. This relates to the first comment about clarifying your epistemological position and using appropriate language,	We agree that the language we use here is not right. We revisited the transcripts as a checking process, to explore how the teams interpreted (or did not interpret) the data. We have amended the description to mention interpret rather than 'identify' and we have removed the 'returning to the original data' text as we do not feel it is necessary.	Text amended lines 311-313.

3	13min	13. Table 3 is slightly confusing: are the numbers hours? Some of the numbers don't add up (eg line 1 for TA team: 11 + 10 = 21, not 20.5).	We apologise for the omission and error in the table. We have amended the table.	See Table 3
3	14min	14. Line 383 – can you explain what you mean by 'polishing content and language	Thank you for highlighting the need for more explanation here.	Additional explanation added lines 440-443.

VERSION 2 – REVIEW

REVIEWER	Sara Garfield Imperial College Healthcare NHS Trust
REVIEW RETURNED	04-Jun-2018

GENERAL COMMENTS	Many thanks for your revisions and clarifications.
--